# Herpesvirus Entry Mediator as an Immune Checkpoint Target and a Potential Prognostic Biomarker in Myeloid and Lymphoid Leukemia

**DOI:** 10.3390/biom14050523

**Published:** 2024-04-27

**Authors:** Fatemah S. Basingab, Reem A. Alzahrani, Aisha A. Alrofaidi, Ahmed S. Barefah, Rawan M. Hammad, Hadil M. Alahdal, Jehan S. Alrahimi, Kawther A. Zaher, Ali H. Algiraigri, Mai M. El-Daly, Saleh A. Alkarim, Alia M. Aldahlawi

**Affiliations:** 1Department of Biological Sciences, Faculty of Science, King Abdulaziz University, Jeddah 21859, Saudi Arabia; 2Immunology Unit, King Fahd Medical Research Center, King Abdulaziz University, Jeddah 21859, Saudi Arabia; 3Hematology Department, Faculty of Medicine, King Abdulaziz University Hospital, King Abdulaziz University, Jeddah 21859, Saudi Arabia; 4Hematology Research Unit, King Fahd Medical Research Center, King Abdulaziz University, Jeddah 21859, Saudi Arabia; 5Department of Biology, Faculty of Science, Princes Nourah bint Abdulrahman University, Riyadh 11671, Saudi Arabia; 6Special Infectious Agents Unit-BSL3, King Fahd Medical Research Center, King Abdulaziz University, Jeddah 21859, Saudi Arabia; 7Department of Medical Laboratory Sciences, Faculty of Applied Medical Sciences, King Abdulaziz University, Jeddah 21859, Saudi Arabia; 8Embryonic Stem Cells Research Unit and Embryonic and Cancer Stem Cells Research Group, King Fahd Medical Research Center, King Abdulaziz University, Jeddah 21859, Saudi Arabia

**Keywords:** herpesvirus entry mediator, acute lymphocytic leukemia, co-inhibitory molecules, immune checkpoint blockade, CD4^+^ T cells, immune checkpoint receptor

## Abstract

Herpesvirus entry mediator (HVEM) is a molecular switch that can modulate immune responses against cancer. The significance of HVEM as an immune checkpoint target and a potential prognostic biomarker in malignancies is still controversial. This study aims to determine whether HVEM is an immune checkpoint target with inhibitory effects on anti-tumor CD4^+^ T cell responses in vitro and whether *HVEM* gene expression is dysregulated in patients with acute lymphocytic leukemia (ALL). *HVEM* gene expression in tumor cell lines and peripheral blood mononuclear cells (PBMCs) from ALL patients and healthy controls was measured using reverse transcription-quantitative polymerase chain reaction (RT-qPCR). Tumor cells were left untreated (control) or were treated with an HVEM blocker before co-culturing with CD4^+^ T cells in vitro in a carboxyfluorescein succinimidyl ester (CFSE)-dependent proliferation assay. *HVEM* expression was upregulated in the chronic myelogenous leukemia cell line (K562) (FC = 376.3, *p* = 0.086) compared with normal embryonic kidney cells (Hek293). CD4^+^ T cell proliferation was significantly increased in the HVEM blocker-treated K562 cells (*p* = 0.0033). Significant *HVEM* differences were detected in ALL PBMCs compared with the controls, and these were associated with newly diagnosed ALL (*p* = 0.0011) and relapsed/refractory (*p* = 0.0051) B cell ALL (*p* = 0.0039) patients. A significant differentiation between malignant ALL and the controls was observed in a receiver operating characteristic (ROC) curve analysis with AUC = 0.78 ± 0.092 (*p* = 0.014). These results indicate that HVEM is an inhibitory molecule that may serve as a target for immunotherapy and a potential ALL biomarker.

## 1. Introduction

A breakthrough in immunotherapy against cancers has occurred with the invention of immune checkpoint blockade (ICB). ICB-based therapy can revive anti-tumor immune responses. The success of ICB in cancer treatment relies primarily on programmed cell death protein 1 (PD-1) and cytotoxic T-lymphocyte-associated protein 4 (CTLA-4) [1]. However, limited response rates of cancer patients to PD-1 and CTLA-4 blockade have been reported [2]. The effectiveness of these blockades is seen only with PD-1^+^ and CTLA-4^+^ cancer patients. In addition, complete resistance to ICB-based therapy has been exhibited in different types of cancer [3]. Therefore, the identification of a new immune checkpoint target in ICB-based cancer therapy is required.

Herpesvirus entry mediator (HVEM) and its ligand B- and T-lymphocyte attenuator (BTLA) have been proposed as promising future targets for cancer immunotherapy. HVEM is a tumor necrosis factor receptor (TNFR) superfamily member associated with the expanding family of immunological checkpoint molecules [4]. It can be a co-stimulatory molecule upon engagement with TNFR LIGHT or lymphotoxin-α [5,6], or a suppressive molecule through interaction with the Ig superfamily (BTLA) [7,8]. Hence, HVEM is described as a molecular switch based on the associated ligand [9,10]. HVEM is expressed in various host cells and contributes to immune homeostasis. It has a broader expression than the well-known PD-1, which is approved for cancer immunotherapy targets [11]. Further, peripheral blood mononuclear cells (PBMCs) of breast cancer patients express the *HVEM* gene in similar levels to the approved target immunotherapy CTLA-4 [12]. 

In cancer, HVEM upregulation has been reported in various tumor cell lines and clinical tumor tissue, including breast and esophageal cancer, chronic lymphocytic leukemia, and melanoma [7,13,14,15]. *HVEM* gene expression is proposed to be a promising prognostic marker in metastatic melanomas when expressed on melanoma cells to interact with BTLA on tumor-infiltrating lymphocytes (TILs) [11]. In myeloid and lymphoid disorders, one of seven acute myeloid leukemia (AML) cell lines exhibits *HVEM* expression. Leukemias of myeloid origin have been commonly classified as having low levels of *HVEM* expression [16,17]. The transformation of chronic myeloid leukemia (CML) into acute lymphoid leukemia (ALL) has been reported in 20–30% of cancer patients during the terminal blast crisis stage, in which more than 20% blasts are present in the patients’ peripheral blood and bone marrow [18].

HVEM has been shown to participate in tumor progression and the evasion of immunosurveillance. For example, HVEM expression is linked to reducing tumor-infiltrating lymphocytes (TILs), affecting anti-tumor immune responses [11]. Anti-HVEM monoclonal antibody (anti-HVEM18-10), which preferentially blocks HVEM engagement with BTLA, has improved the immune responses of γδ-T cells against lymphoma [19,20]. Administering anti-HVEM18-10 in prostate tumor-bearing mice in vivo showed a twofold reduction in tumor growth and reconstituted human T cells [21]. Nevertheless, no previous study has examined HVEM blocker in leukemia. Instead, an anti-BTLA monoclonal antibody has been shown to deplete chronic lymphocytic leukemia (CLL) and boost natural killer (NK) cell-mediated responses ex vivo by increasing IFN-γ production and cytotoxic capability in antibody-dependent cytotoxicity (ADCC). Although BTLA is upregulated in CLL, HVEM is downregulated in both NK and leukemic cells in patients with CLL associated with advanced Rai/Binet stage [22]. Other researchers, however, found HVEM to enhance both tumor regression and anti-tumor responses. Investigations have demonstrated that HVEM overexpression increases the survival incidence of pancreatic and bladder tumors [23,24,25]. Chemotherapy with adoptive cells using CD8^+^ BTLA^+^ TILs results in better clinical outcomes for managing melanoma through an increased response to IL-2. The ligation of BTLA with HVEM has been shown to activate the Akt/PKB pathway, thus preventing the apoptosis of CD8^+^ BTLA^+^ TILs [26]. Two important themes that emerge from the studies discussed are that HVEM either encourages tumor progression by inhibiting anti-tumor immune responses or favors tumor regression by enhancing anti-tumor immunity. The extent to which HVEM acts in leukemia remains uncertain. 

Numerous studies have focused on acute myeloid leukemia (AML); however, only a few have examined acute lymphocytic leukemia (ALL). The use of HVEM blockade against leukemia has yet to be investigated. The clinical significance of HVEM as an immune checkpoint target with a potential prognostic marker in malignant tumors has been hypothesized in various cancers, but not in ALL. Therefore, the aim of this study is to determine whether HVEM has an inhibitory effect on anti-tumor CD4^+^ T cell responses in vitro and whether *HVEM* gene expression is dysregulated in patients with ALL. We hypothesize that tumor-expressing HVEM can inhibit the proliferation of CD4^+^ T cells in vitro and that *HVEM* gene expression is a prognostic biomarker upregulated in ALL. 

## 2. Materials and Methods

### 2.1. Study Design

To determine whether HVEM has an inhibitory effect on anti-tumor CD4^+^ T cell responses in vitro, an experimental approach was employed in three stages. First, the expression of the *HVEM* gene and surface protein were measured in various tumor cell lines in vitro, resulting in the selection of tumor cell lines that expressed the highest *HVEM*. CD4^+^ T cells were then isolated from healthy donors. Last, the HVEM^+^ tumor cells were treated with HVEM blockade before being co-cultured with CD4^+^ T cells. The proliferation of CD4^+^ T cells in response to HVEM blockade-treated and untreated tumor cells was measured via a CFSE-dependent assay.

To ascertain whether HVEM gene expression is dysregulated in patients with ALL, reverse transcription-quantitative polymerase chain reaction (RT-qPCR) was utilized on RNA isolated from malignant ALL patients and non-malignant healthy controls. Correlations between HVEM gene expression and clinicopathological data were conducted. Further, receiver operating characteristic (ROC) curve analysis was performed to evaluate the diagnostic ability of HVEM gene expression to discriminate between malignant ALL and non-malignant healthy controls. 

### 2.2. In Vitro Tumor and Healthy Control Cell Lines

To examine the effect of HVEM-expressing tumor cells on T cell proliferation in vitro, it is important to determine the expression of HVEM in different tumor cell lines. Variations in gene expression have been detected between an adherent and a non-adherent cell suspension of Hek293 after transcriptomic, genomic, and metabolic gene analysis, which are associated with cell membrane proteins [27]. As HVEM is a surface protein, two adherent tumor cell lines, breast cancer (MCF-7) and hepatocellular carcinomas (HepG2) and non-adherent chronic myelogenous leukemia (CML) (K562), were also utilized. An embryonic immobilized kidney cell line (Hek293) served as a healthy control [28]. All cell lines were obtained from the Immunology Unit at King Fahd Medical Research Center (KFMRC). Tumor cells were grown, and passaged in RPMI media, as previously described [29]. All cell lines were propagated in RPMI-1640 supplemented with 10% FBS, 100 units/mL penicillin, and 100 µg/mL streptomycin at 37 °C in a 5% CO_2_ incubator. 

### 2.3. CFSE CD4^+^ T Cell Proliferation in Response to HVEM Blocker-Treated and Untreated Tumor Cells In Vitro

To examine the proliferation of CD4^+^ T cells in response to HVEM^+^ tumor cells, CD4^+^ T cells were isolated from the peripheral blood of healthy individuals by the Roaya Unit at King Fahd Medical Research Center (KFMRC). Blood samples were collected in ethylenediaminetetraacetic acid (EDTA) tubes (BRT; Qiagen, Inc., Manchester, UK). PBMCs were prepared as previously shown in a published article [29]. An amount of 14–16 mL of a whole blood sample with approximately 15.7 × 10^6^ PBMC cells was collected using Lymphoprep density gradient medium (Stem Cell Technologies, Vancouver, BC, Canada), then CD4^+^ T cells were purified from PBMCs using CD4 Miltenyi beads (Miltenyi Biotec, Gaithersburg, MD, USA) following the manufacturer’s instructions. Purified CD4^+^ T cells were labeled with carboxyfluorescein succinimidyl ester (CFSE: BioLegend, San Diego, CA, USA) in the dark for 20 min at room temperature according to the instructions before being washed and used in co-culturing with tumor cells [30]. 

K562 and Hek293 cells were either treated or left untreated with 20 ng/100 µL of HVEM-TNFRSF14 antibody blocking peptide (NBP1-76690PEP) (Novus Biologicals, Centennial, CO, USA). The peptide product is purified and corresponds to the immunogen sequence of the corresponding antibody. HVEM blocker was prepared according to the manufacturer’s instructions and added to 1 × 10^5^ of either K562 or Hek293 for 2 h before being washed and seeded in two separate six-well plates. Next, 1 × 10^6^ CFSE-labeled CD4^+^ T cells were added. The plates were incubated for 72 h at 37 °C in a 5% CO_2_ incubator [29,30,31].

### 2.4. Flow Cytometry

To determine the purity of the CD4^+^ T cells isolated from PBMCs, the CD4^+^ T cells were stained with anti-CD4 and anti-CD8 monoclonal antibodies (mAbs) (Bio-Legend, San Diego, CA, USA). Purified CD4^+^ T cells, collected after MACS separation or at the end of co-culturing with tumor cells, were treated with Fc blocker at room temperature for 15 min. Next, the T cells were stained with anti-CD4-PE-Cy7 and anti-CD8-APC-Cy7 mAbs (BioLegend, USA) for 30 min at 4 °C before being washed. The CD4^+^ T cells were then stained with 7AAD to exclude any dead cells. The tumor cells were stained with anti-HVEM-PE (Bio-Legend, USA) to ascertain the levels of HVEM surface protein. The stained cells were run on BD FACSAriaTM III flow cytometry (Becton Dickinson, Franklin Lakes, NJ, USA) according to published studies [29,32]. Data were then analyzed using FlowJo^TM^ version 10 software (Becton Dickinson, USA). 

### 2.5. Study Subjects

Blood samples from 23 patients with acute lymphocytic leukemia (ALL) and 10 non-malignant healthy controls were obtained between 2021 and 2023 from the Department of Hematology at King Abdulaziz University Hospital (KAUH), Jeddah, Saudi Arabia. ALL blood samples were collected from the following: 1. newly diagnosed patients before starting treatment, 2. patients in the remission phase, and 3. patients in the relapse/refractory phase. All participants or legal guardians were informed of the objective of the study and were required to sign a consent form before participating in the study. An amount of 3–5 mL of peripheral blood samples was collected into EDTA tubes using PAXgene™ blood RNA tubes (BRT; Qiagen, Inc.) according to the manufacturer’s instructions [12]. The collected blood samples were stored at −80 °C and used for RNA extraction. Ethical approval number HA-02-J-008 (Reference No. 512/21) was granted from the Biomedical Ethics Research Committee of KAUH, Jeddah, Saudi Arabia. 

### 2.6. Reverse Transcription-Quantitative PCR (RT-qPCR)

To measure *HVEM* gene expression from the tumor cells and blood samples, total RNA was extracted from the tumor cultures using RNAbler Cells and Tissue Kit (Haven Scientific, Thuwal, Saudi Arabia) and from blood samples using (Qiagen, Inc., UK), as per the manufacturer’s instructions. The concentration and purity of the RNA samples were checked using a NanoDrop 2000c spectrophotometer (Thermo Fisher Scientific, Waltham, MA, USA) at a 260/280 ratio ~2 and a 260/230 ratio ~2.2. All steps in this protocol were carried out at room temperature.

A QuantiTect Reverse Transcription Kit (Qiagen, Inc.) was utilized to prepare cDNA from the required RNA template of interest, as per the manufacturer’s protocol. The cDNA product was stored at −20 °C for downstream gene expression analysis.

RT-qPCR was used to evaluate the expression levels of *HVEM* in three tumor cell lines and a healthy control sample. Primers targeting these genes were designed using NCBI gene databases and the Primer3Web tool. The primer sequence was as follows: 

HVEM: (F) acttctgcatcgtccaggac, (R) tctggtgctgacattcctcc,

GAPDH: (F) agaacgggaagcttgtcatc, (R) ggcagagatgatgacccttt.

The relative gene expression levels were adjusted using the internal reference housekeeping gene glyceraldehyde 3-phosphate dehydrogenase (GAPDH). The samples were processed in duplicate in a 96-well plate using BioFact™ 2X Real-Time PCR Master Mix SYBR Green and a Real-Time PCR device (Thermo Fisher Scientific, Inc.) according to the manufacturer’s instructions. RT-qPCR was conducted using a single initial cycle of 30 s at 95 °C, followed by 40 amplification cycles of 15 s at 98 °C and 30 s at 60 °C. The amplified products were verified at the end of each cycle, and their purity was determined by analyzing the melting curves. Relative expression was quantified using the comparative Ct method (2^−ΔΔCt^) [33,34]. 

### 2.7. Statistical Analysis

IBM SPSS Statistics 27.0.1.0 software and GraphPad Prism 10.0.3 (GraphPad Software, Inc., La Jolla, CA, USA) were used for statistical analyses of the relative gene expression of *HVEM*, where *p* ≤ 0.05 was considered statistically significant. Non-parametric tests were chosen based on whether the included variables are categorical or have a skewed distribution, accordingly. Any significant changes in gene expression between non-malignant controls and ALL samples were noted using an unpaired two-tailed *t*-test. Additionally, a one-way ANOVA (two-tailed Mann–Whitney and Kruskal–Wallis tests) was applied to select the parameters to compare among the three groups. The results are presented as the mean ± standard error of the mean (SEM). To examine the sensitivity and specificity of *HVEM* as a possible biomarker, receiver operating characteristic (ROC) curves were plotted using their expression values (2^−ΔΔCq^) in malignant ALL and non-malignant controls. *p* ≤ 0.05 was considered to indicate a statistically significant difference [30,35]. 

## 3. Results

### 3.1. HVEM Gene Expression and HVEM Surface Protein Are Upregulated in Different Cancer Cell Lines In Vitro

To examine the effect of HVEM blocker on HVEM-expressing tumor cells and on the proliferation of T cells in vitro, the expression of HVEM was measured in different tumor cells. In this study, the following three different tumor cell lines were utilized: breast cancer (MCF-7), hepatocellular carcinomas (HepG2), and chronic myelogenous leukemia (CML) (K562), along with human embryonic kidney (Hek293), which served as a healthy control indicator. MCF-7, HepG2, and Hek293 are adherent cells, whereas K562 are cells that grow in suspension. The cell morphology and HVEM expression at the gene and surface protein levels were investigated. Under high-magnification light microscopy, MCF-7 cells appeared elongated with a large and centered nucleus compared with HepG2 cells, which are epithelial-like cells with sharp ends. K562 leukemia cells have a shiny rounded shape, and Hek293 cells are spheroidal in shape (Figure 1). The results in Table 1 show an upregulation of *HVEM* gene expression in the three cancer cell lines, MCF-7, HepG2, and K562, compared with the healthy control Hek293 cells with fold changes (FCs) of 1.8, 2.3, and 376, respectively. Leukemic cell line K562 showed the highest degree of upregulation of *HVEM* with an FC of 376 (*p* = 0.086) compared with the healthy control Hek293. In addition, HVEM surface protein expression was measured using flow cytometry. The surface expression of HVEM protein is represented as the percentage of HVEM^+^ cells and the mean of fluorescence intensity (MFI). The percentages of HVEM-positive cells were around 8%, 21%, and 11% of MCF-7, HepG2, and K562, respectively, compared with 1% of HVEM^+^ on Hek293 cells. The MFI equaled 56, 62, and 146 on MCF-7, HepG2, and K562, respectively, compared with their unstained controls, and an MFI of 40 was detected on Hek293. Although 1% of K562 was HVEM^+^ cells, there was a distinct separation between the unstained and HVEM-stained cells, and the highest HVEM MFI was detected on K562. Based on these findings, K562 and Hek293 cells were chosen for further investigations.

### 3.2. CD4^+^ T Cell Proliferation in Response to HVEM Blocker-Treated and Untreated Tumor Cells In Vitro

To examine the effect of blocking HVEM expressed by K562 cells on the proliferation of T cells in vitro, it was essential to purify the CD4^+^ T cells from the PBMCs obtained from the blood samples of healthy donors using MACS CD4 beads. Following the CD4 bead enrichment process, 6.72 × 10^6^ CD4^+^ T cells were collected, representing approximately 42% of the PBMCs. Similar results were observed based on the flow cytometric analysis data with anti-CD4^+^ and anti-CD8^+^ mAbs, which showed that CD4^+^ T cells represented 39–45% of the PBMCs and the CD4^+^ purity yield reached around 80% post-selection (Appendix A). Purified CD4^+^ T cells were then labeled with CFSE, with a loss of CFSE indicating the division of T cells. 

HVEM^+^ K562 cells and HVEM^−^ Hek293 were treated first with 20 ng/100 μL of HVEM blocker using 1 × 10^5^ cells for 2 h before being washed and then co-cultured with CFSE-labeled CD4^+^ T cells at a ratio of one tumor or healthy control cell to ten CFSE CD4^+^ T cells. The results showed that HVEM^+^ K562 tumor cells expanded normally when they were cultured alone or co-cultured with CFSE CD4^+^ T cells without HVEM blocker treatment of the tumor cells. In contrast, the number of tumor cells reduced significantly when HVEM^+^ K562 cells were treated with the HVEM blocker before being co-cultured with CFSE CD4^+^ T cells (Figure 2, left column). In the Hek293 cultures, the cells appeared similar when the Hek293 cells were grown alone or with CFSE CD4^+^ T cells post-Hek293 cell treatment with HVEM blocker (Figure 2, right column). These results indicated that the HVEM blocker at the used concentration of 20 ng/100 μL was not toxic because the healthy control Hek293 cells grew normally after HVEM blocker treatment. The reduction in HVEM^+^ K562 cells was investigated further using flow cytometric analysis. 

The proliferation of CFSE CD4^+^ T cells in response to HVEM blocker-treated and untreated HVEM^+^ K562 or HVEM^−^ Hek293 is shown in Figure 3. Approximately 11% of CFSE CD4^+^ T cells reached the fifth division after 72 h of incubation when the CD4^+^ T cells were cultured with HVEM^+^ K562. Treating HVEM^+^ K562 with an HVEM blocker before co-culturing increased the loss of CFSE, and approximately 20% of CFSE CD4^+^ T cells reached the fifth division, indicating that blocking HVEM enhanced the T cell proliferation (Figure 3A, bottom). No differences were reported in the CFSE CD4^+^ T cell proliferation when co-cultured with HVEM blocker-treated or untreated HVEM^−^ Hek293 (Figure 3B, bottom). 

For cell viability, 75% of HVEM^+^ K562 tumor cells remained alive after co-culturing with CFSE CD4^+^ T cells as a result of not treating them with an HVEM blocker. However, only 10% of the same cells remained alive after co-culturing with CFSE CD4^+^ T cells upon the treatment of cells with an HVEM blocker before co-culturing (Figure 3A, top). This reduction in cell viability was not detected on HVEM^−^ Hek293 where the cells remained alive after co-culturing with CFSE CD4^+^ T cells in both HVEM blocker-treated and untreated Hek293 cells (Figure 3B, top). 

To validate the proliferation of CFSE CD4^+^ T cells results, the mean fluorescence intensity (MFI) of CFSE was measured in the CD4^+^ T cells when co-cultured with the K562 or Hek293 cells that had been either left untreated or treated with 20 ng/µL of HVEM blocker. There was a significant reduction in the MFI of CFSE CD4^+^ T cells with the HVEM blocker-treated K562 (MFI = 116.5 ± 1.18) compared with the HVEM blocker untreated K562 (MFI = 136 ± 1.18) with *p* = 0.0033, indicating a higher proliferation of CD4^+^ T cells in response to HVEM blocker-treated K562 cells. No significant differences were detected in the co-culturing with HVEM blocker-treated and untreated healthy control Hek293 (Figure 4).

### 3.3. Elevated HVEM Gene Expression in Acute Lymphocytic Leukemia (ALL)

To measure the gene expression of *HVEM* in the PBMCs of cancer patients, 23 blood samples were collected from ALL patients and compared with 10 non-malignant healthy controls. Table 2 shows the baseline characteristics of the ALL and healthy control subjects. The results show a significantly higher expression of *HVEM* among the ALL patients compared with the non-malignant healthy controls (*p* = 0.0064) (Figure 5A). In addition, significance was noted among the newly diagnosed ALL patients and those in the relapse/refractory stage (*p* = 0.0011 and *p* = 0.0051, respectively) (Figure 5B). Moreover, the significant differences in *HVEM* expression between the non-malignant controls and the ALL patients are associated with the B-ALL type (*p* = 0.0039). In contrast, no significant *HVEM* expression differences were reported between the non-malignant controls and the pre-B-ALL or T-ALL patients (Figure 5C). A ROC curve analysis showed that *HVEM* expression allowed significant differences between the patients with malignant ALL and the non-malignant healthy controls, with an area under the curve (AUC) equal to 0.78 ± 0.092 (*p* = 0.014) (Figure 6A). In addition, a significant difference value of AUC = 0.89 ± 0.088 (*p* = 0.013) was calculated among the patients with newly diagnosed ALL and the non-malignant healthy controls (Figure 6B). Unlike with the ALL patients in the remission phase, the discriminatory power of the AUC test is of limited diagnostic value, with an AUC value equal to 0.65 ± 0.111, indicating non-significance (*p* = 0.198) (Figure 6C). Moreover, an AUC value of 0.78 ± 0.098 (*p* = 0.036) was obtained among the ALL patients in the relapse/refractory phase and the non-malignant healthy controls (Figure 6D). This indicates perfect diagnostic accuracy, which is clinically useful, and the result is highly significant (*p* = 0.04). These results suggest that *HVEM* at the gene levels may act as a potential biomarker for malignant ALL in the newly diagnosed and relapsed/refractory phases. 

## 4. Discussion

Cancer immunotherapy has advanced rapidly with the inclusion of numerous ICBs approved against specific targets, such as ipilimumab, an inhibitor of CTLA-4, and nivolumab and pembrolizumab, PD-1 inhibitors [3,36,37]. However, there are limitations associated with these novel drugs, mainly adverse effects that emerge from systemic host immunosuppression, along with the resistance mechanism developed by the host immunity. In addition, such ICBs are efficient for use with the subset of cancer patients who are CTLA-4-positive and PD-1-positive [22]. 

HVEM is an immune checkpoint protein that, upon ligation with BTLA, regulates the T cell immune response. Abnormal HVEM expression has been detected not only in various host immune cells [9], but also in several types of cancers [13,23,24]. *HVEM* expression is elevated in hepatocarcinoma [38], breast cancers [39], chronic lymphocytic leukemia (CLL) [13], and prostate cancer [28]. In this study, the significant upregulation of *HVEM* expression in leukemic cell lines in vitro at the gene and protein levels corroborated these earlier findings. These results differ from Duhen et al. (2004), who reported HVEM downregulation in leukemic cells in patients with CLL; however, those results were associated with advanced Rai/Binet stage [6]. The current study also found that higher *HVEM* mRNA expression was associated with non-adherent leukemic cells, compared with adherent breast cancer and hepatocellular carcinomas and healthy human embryonic kidney (Hek293). In agreement with our study, previous studies have shown that *HVEM* is expressed by A20 leukemia cells [40]. The variation in surface protein expression between adherent and non-adherent Hek293 cells observed in an earlier study was associated with cellular components, motility, and adhesion molecules [27]. However, no evidence relating to the differences in *HVEM* expression between adherent and non-adherent cells was reported. Although Hek293 cells are immortalized human embryonic kidney cells, they were used as a healthy control in our study because they showed zero ∆∆Ct of *HVEM* gene expression. The low *HVEM* expression levels on Hek293 were noted previously because these cells were transfected with recombinant HVEM plasmid to induce the expression of HVEM. Hek293 cells were also used as healthy controls in previous studies [4,41,42,43]. The difficulty of growing ALL in vitro restricted the use of ALL cell lines in testing HVEM blockade in this current study. ALL blasts rely on their in vivo environment and undergo apoptosis ex vivo. Drug sensitivity testing in vitro with ALL is not commonly performed, owing to the fast fall of ALL cells in these experiments, even when no anti-leukemic drugs are used [44]. Mycoplasma contamination occurs in 5–10% of ALL cell lines. Moreover, 16–38% of ALL cell lines cross-contaminate with other ALL cell lines or lose their phenotypic features after several passages. ALL cell lines are not constantly in circulation, and they invade other tissues. As a result, supplying growth factors, adhesive substrates, and cytokines may result in treatment resistance [45] Based on the above reasons, CML (K562), which expresses the highest HVEM levels, was used in the HVEM blockade experiment. 

Herein, the therapeutic target of HVEM was evaluated against K562 cancer cell lines in vitro. The efficacy of blocking HVEM expressed on K562 leukemic cells by NBP1-76690PEP (Novus Biologicals, USA) on the proliferation of naïve CFSE CD4^+^ T cells showed an increase in CD4^+^ T cell proliferation in vitro. Accordingly, anti-HVEM mAbs enhanced γδ-T cell immune responses; however, this was against lymphoma [5,20]. Anti-HVEM 18-18 mAbs improved the activity of primary human αβ-T cells and decreased exhausted CD8^+^ and regulatory T cells [46]. In addition, administering an HVEM blockade in prostate tumor-bearing mice in vivo reduced tumor growth by twofold and reconstituted human T cells [21]. Moreover, inhibiting the interaction of BTLA with HVEM using anti-BTLA mAbs boosted NK cell-mediated responses ex vivo by increasing their IFN-γ production [6]. Many studies have shown that BTLA, an HVEM ligand, is expressed more in naïve T cells than in memory T cells [5,7,47,48]. The levels of BTLA can transiently increase upon T cell differentiation and activation and eventually decrease in activated T cells [6,49]. Although both CD4^+^ and CD8^+^ T cells express BTLA, a ligand of HVEM, CD4^+^ T cells have been reported to express more BTLA than CD8^+^ T cells [8,50]. BTLA increases more significantly in circulating CD4^+^ T cells than in CD8^+^ T cells in hepatocellular carcinoma (HCC) [9,51]. Interestingly, blocking HVEM in this study not only enhanced T cell proliferation but also diminished leukemic cells in vitro. These results reflect those of del Rio et al. (2023), who also found that the expression of HVEM on A20 leukemic cells is essential in maintaining the proliferation and survival of tumor cells because of the deletion of the HVEM-induced PD-1 stem cell-like T cells that contain the tumor progression [40]. HVEM expression has been linked to a reduction in tumor-infiltrating lymphocytes (TILs), thus affecting anti-tumor immune responses in melanoma [11]. The ligation of BTLA with HVEM activates the Akt/PKB pathway, thus preventing the apoptosis of CD8^+^BTLA^+^ TILs, resulting in the enhancement of immune responses [26]. Blocking BTLA/HVEM pathways promotes the production of IFN-γ by circulating T cells [9]. In summary, HVEM contributes to the following three hallmarks of cancer: sustaining proliferative signals, resisting apoptosis, and evading immune distraction [52]. It has been shown that CML can transform into ALL in 20–30% of cancer patients during the terminal blast crisis stage [18]. 

Research to date has not determined whether HVEM gene expression is dysregulated in patients with ALL. In this study, we also identified *HVEM* gene expression as a prognostic factor in ALL patients. Numerous reports have measured the expression of *HVEM* in acute myeloid leukemia (AML) [53,54]. Gene dysregulation was reported in studies investigating *HVEM* expression in patients with AML [54]. Elevated *HVEM* expression is reported to be associated with a worse cancer prognosis and a lower survival rate in cancer patients. The upregulation of *HVEM* was shown to be related to breast cancer aggressiveness [39]. Similarly, a low response and high incidence of relapse were associated with immune checkpoints in AML. This finding is consistent with our study, in which significant *HVEM* upregulation was reported in relapsed and refractory ALL patients. In addition, newly diagnosed ALL patients have upregulated *HVEM* expression. Moreover, we have previously shown a positive correlation between *HVEM* gene expression and HVEM serum levels in breast cancer with a higher tumor grade and a worse cancer prognosis [12,28]. The serum levels of HVEM are elevated compared with serum CTLA-4 in breast cancer PBMCs. These indicate that, although CTLA-4 remains on immune cells, HVEM can be shed from cells; thus, blocking HVEM would target both circulating and surface molecules with border effects. In contrast to the data of the current study, *HVEM* expression has also been reported to improve the prognosis of cholangiocarcinoma (ICC), favoring overall survival [55]. The significant upregulation of *HVEM* is more highly associated with B-ALL compared with T-ALL and non-malignant controls. One study reported that HVEM as a surface protein frequently mutates in germinal center-derived B cell lymphomas. It suggested that the interaction of BTLA on T cells with HVEM on B cells results in BTLA regulating the expression of HVEM on B cells through phosphatase SHP1, resulting in a reduction in T cell receptors (TCRs). Without BTLA on T cells, B cells upregulate Bcl-2, leading to germinal center B cells outgrowing [43]. Another study showed that the expression of HVEM was considerably elevated in αβ^+^ T cells in the low-risk group of acute myeloid leukemia (AML) patients, but CTLA-4 on γδ^+^ T cells and PD-1 ligand on blasts were both correlated with poor relapse-free survival in ALL patients [56]. 

To ensure the validity of the conducted experiments, we employed a rigorous experimental design, appropriate sampling and measurement tools, and statistical analysis. For the experimental design, as in previous studies, all experiments used a single variable and a control. Examples of the controls utilized in this study are Hek293 in vitro cells [4,41,42,43], HVEM untreated tumor cells [46], unstained CD4^+^ T cells [30], and non-malignant healthy controls [35]. In addition, the replication of each experiment was conducted similarly to previous publications [30,35]. Moreover, various programs were employed to analyze data, such as FlowJo for flow cytometric data and GraphPad Prism for statistical analysis [30,35]. The results included in this study indicate whether the differences between groups are significant based on *p*-values [12]. The consistency, stability, and repeatability of the experiment’s results confirm the reliability of our study. 

The current study has some limitations. ALL samples with the three study criteria, newly diagnosed, remission, and relapse/refractory, were difficult to obtain. The number of collected samples was 33, with 23 patient samples and 10 healthy controls. The mean age ± SEM of the healthy control subjects was 35 ± 3, compared with 8 ± 1.3 for the ALL patients. Collecting blood samples from healthy pediatric patients who were not admitted to hospital was a constraint. In the case of collecting samples from hospital-admitted pediatric patients, other underlying health conditions could not be excluded. Future ongoing studies are underway in our laboratory to delineate the HVEM mechanism through the molecular silencing of the HVEM gene and examine the effect on T cell responses. In addition, performing a phenotypic analysis of PBMCs may provide insights into using HVEM as a biomarker in ALL. The serum levels of HVEM will be tested and compared with the *HVEM* gene levels. 

## 5. Conclusions

This study concludes that HVEM expression is upregulated in many types of cancer, including CML and ALL, and HVEM protein acts as an immune suppressive molecule that contributes to tumor progression and may serve as a potential biomarker and target for cancer immunotherapy.

## Figures and Tables

**Figure 1 biomolecules-14-00523-f001:**
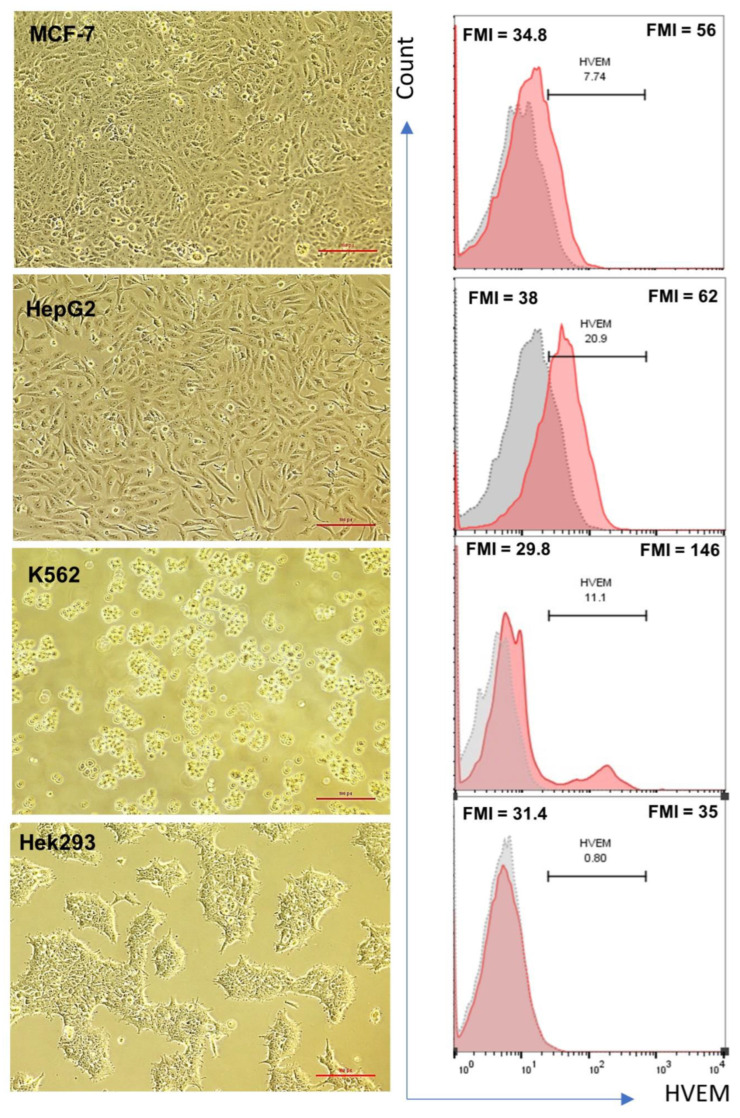
Light microscopy images and the expression of herpesvirus entry mediator (HVEM) surface protein on breast cancer (MCF-7), hepatocellular carcinomas (HepG2), chronic myelogenous leukemia (K562), and human embryonic kidney (Hek293) cells. The left column represents the microscope and a magnification of 100 pt using NIS-Elements F 4.00.00 software (Nikon Instruments, USA) under a light microscope (Nikon Eclipse, Fujisawa, Japan). Data collected from four separate experiments represents cell proliferation, morphology, and attachment nature. The right column shows histogram graphs representing the measurements of HVEM expression after staining with anti-HVEM-PE. The dotted-line histograms show the unstained control cells, whereas the line histograms represent the HVEM-stained cells. The numbers at the top of each histogram indicate the MFI of the unstained (**left**) and HVEM-PE (**right**). The number in the middle represents the percentage of HVEM-positive cells. The results were collected from two wells in three separate experiments. Magnification: 100 pt.

**Figure 2 biomolecules-14-00523-f002:**
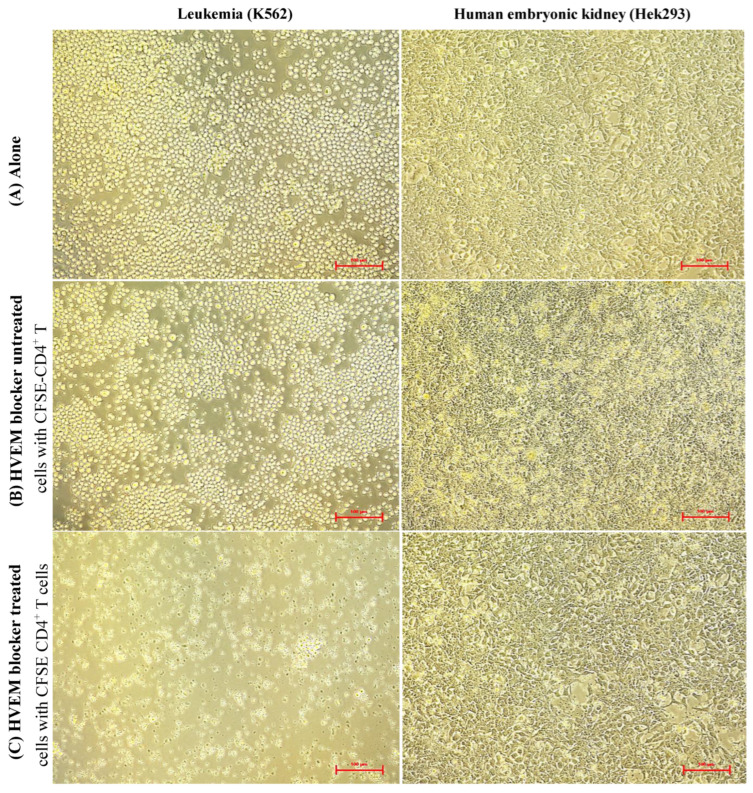
Light microscopic images of K562 (**left**) and Hek293 (**right**): (**A**) alone, (**B**) HVEM untreated cells with CFSE CD4^+^ T cells at one tumor or normal to ten T cells, or (**C**) HVEM blocker treated cells with CFSE CD4^+^ T cells after K562 and Hek293 for 72 h of incubation. Data are representative of three separate experiments. Magnification: 100 μm.

**Figure 3 biomolecules-14-00523-f003:**
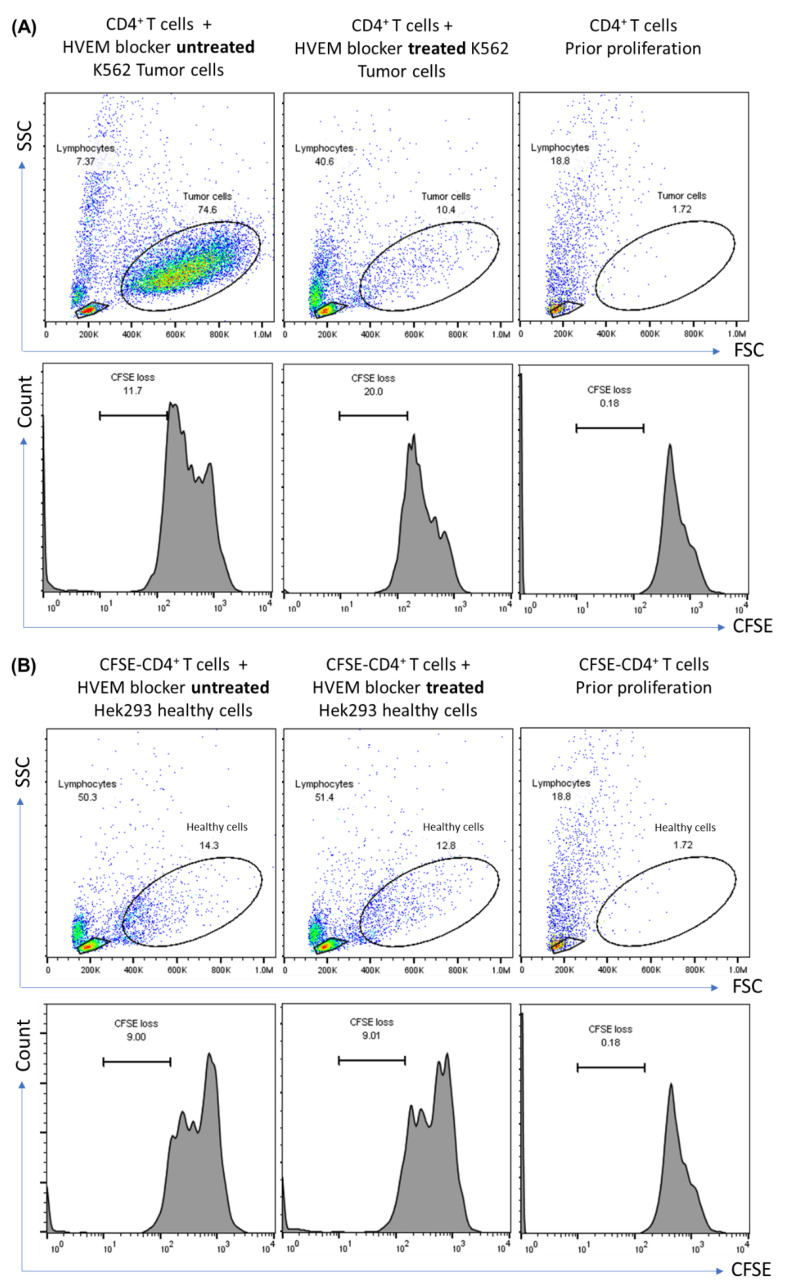
Proliferation of CFSE-labeled CD4^+^ T cells in response to (**A**) K562 tumor cells or (**B**) Hek293 healthy control cells without HVEM blocker treatment (**left** column), or after K562 and Hek293 treatment with HVEM blocker (**middle** column); and non-proliferated CFSE CD4^+^ T cells (**right** column) after 72 h of incubation at 37 °C. Data are representative of two separate experiments.

**Figure 4 biomolecules-14-00523-f004:**
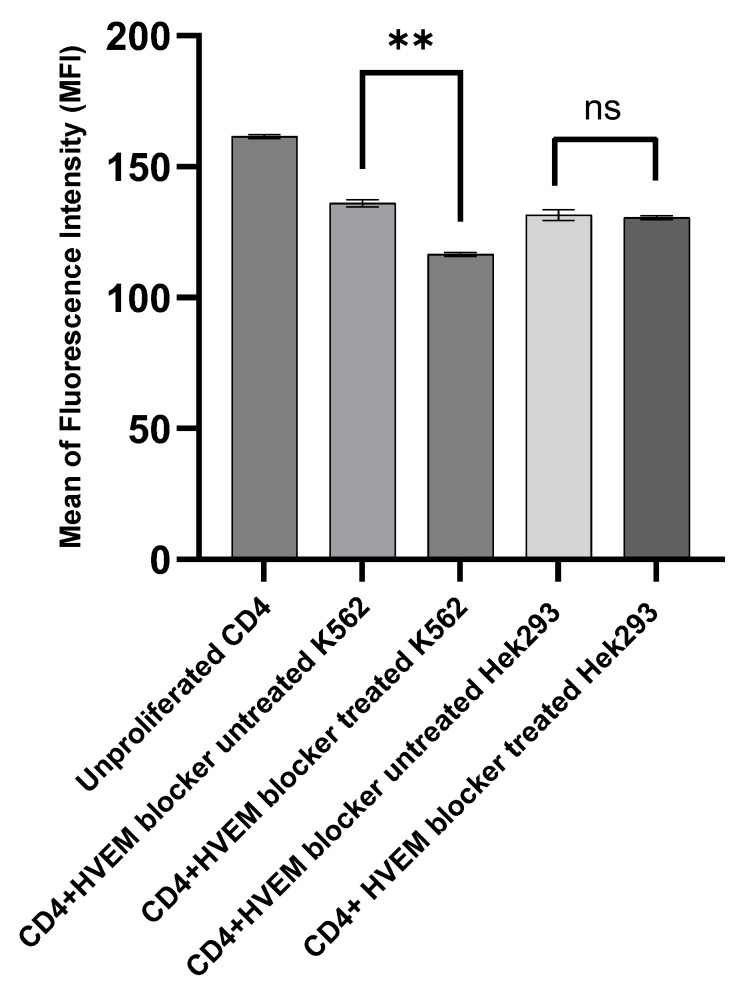
Mean ± SD of fluorescence intensity of CFSE-labeled CD4^+^ T cells before proliferation on day zero and after 72 h of incubation with either K562 tumor cells or with Hek293 normal cells that were left untreated or treated with 20 ng HVEM blocker. Cells were cultured at a ratio of one tumor or normal cell to ten CFSE CD4^+^ T cells. ** indicates strong significant differences between groups with *p* = 0.0033, whereas “ns” indicates no significant differences with *p* = 0.5918. The results were collected from three separate experiments.

**Figure 5 biomolecules-14-00523-f005:**
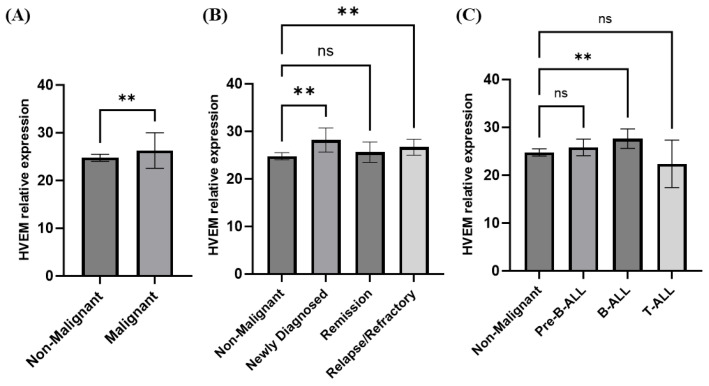
Relative expression of *HVEM* gene in patients with malignant ALL compared with non-malignant healthy controls. RNA from PBMCs were isolated, RT-qPCR determined the gene expression of HVEM, and the expression of GAPDH was normalized. (**A**) represents HVEM expression in all study subjects, 10 non-malignant controls and 23 malignant ALL patients, and ** shows significant differences between groups with *p* = 0.0064 (using the Mann–Whitney multiple comparisons test). (**B**) shows the expression of HVEM in three ALL categories (newly diagnosed, remission, and relapse/refractory) compared with the non-malignant control, and ** shows significant differences between non-malignant and either freshly diagnosed groups with *p* = 0.0011 or relapse/refractory with *p* = 0.0051 (using a one-way ANOVA multiple comparison test). (**C**) represents HVEM relative expression in three types of ALL (pre-B-ALL, B-ALL, and T-ALL) compared with the non-malignant controls, and ** shows significant differences between groups with *p* = 0.0039 (using a one-way ANOVA multiple comparison test). In addition, “ns” represents non-significant between groups. All statistical analyses were conducted using GraphPad Prism 10.0.3.

**Figure 6 biomolecules-14-00523-f006:**
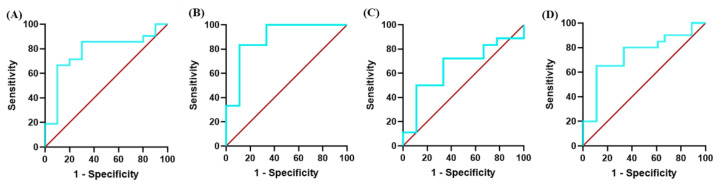
Receiver operating characteristic curve for *HVEM* gene expression in patients with malignant ALL compared with non-malignant healthy controls: (**A**) in all samples, (**B**) in newly diagnosed ALL, (**C**) in remission ALL, and (**D**) in relapse/refractory. AUC, the area under the curve, values are 0.7762 (*p* = 0.0142), 0.8889 (*p* = 0.0133), 0.6543 (*p* = 0.1985), and 0.7472 (*p* = 0.0359), respectively, indicating that *HVEM* is a potential biomarker in ALL patients, especially in newly diagnosed patients and those in the relapse/refractory phase.

**Table 1 biomolecules-14-00523-t001:** The relative gene expression level of *HVEM*, normalized to the housekeeping gene *GAPDH*, is represented as a fold change.

Cell Type	*HVEM* Ct Mean	*GAPDH* Ct Mean	∆Ct	∆∆Ct	Fold Change (FC)
MCF-7	32.486	21.885	10.602	−0.854	1.8
HepG2	33.60	23.223	10.230	−1.225	2.3
K562	25.268	22.368	2.900	−8.556	376.3 *
Hek293	33.230	21.774	11.456	0	1

A non-parametric pair-wise comparison among the four cell line groups was tested using SPSS software, and a biologically significant increase in gene expression was determined between K562 and Hek293 (* indicates significancy, *p* = 0.086).

**Table 2 biomolecules-14-00523-t002:** Baseline characteristics of acute lymphocytic leukemia (ALL) and non-malignant healthy control subjects.

	Total			Non-Malignant			Malignant			
Parameters	Mean ± SEM	Median	IQR	Mean ± SEM	Median	IQR	Mean ± SEM	Median		IQR
Number of participants, n (%)	33 (100)			10 (30)			23 (70)			
Gender	-			6 Female (60%), 4 Male (40%)			13 Female (57%), 10 Male (43%)			
Age (years)	17 ± 3	12	21.25	35 ± 3	38	12.5	8 ± 1.3	7		4
Initial WBC count (unit)	-	-	-	-	-	-	74.13 ± 39	13.3		54.65
Disease status at study enrollment										
Newly diagnosed			Remission		Relapse		Refractory			
3 (13%)			8 (35%)		10% (45%)		5 (22%)			
Male	Female		Male	Female	Male	Female	Male		Female	
2	1		3	5	4	6	2		3	

## Data Availability

Not applicable.

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
