# Peer review of "Herpesvirus Entry Mediator as an Immune Checkpoint Target and a Potential Prognostic Biomarker in Myeloid and Lymphoid Leukemia"

_biomolecules, 2024, doi:10.3390/biom14050523_

Round 1

Reviewer 1 Report

Comments and Suggestions for Authors

Thank you for inviting me to review the above-titled manuscript. The topic is interesting. However, there are problems in the manuscript.

Abstract- 1) I'm not sure why these numbers? Is this the journal style? 2) What is the research question? 3) What are you trying to prove? 4) Methods- What are you testing? 5) what samples? 6) Are these in-vitro or in-vivo studies? What is the study design? How did you assess your data? 7) Results cannot prove a cause-and-effect. It can be an association. No strong evidence for your conclusions. The last line in the conclusion should be omitted.

Introduction- 1) The introduction should be focused and show related studies. 2) What is the research question? What are you trying to prove? 

Methods- 1) Needs citations and references in different sections. What are the references for the method? 2) What is the validity and reliability? 3) The "newly diagnosed patients"- Did you start them on treatment? or the research started before their treatment? 3) "Blood samples were collected by the Roaya Unit at King Fahd Medical Research Cen-102 ter (KFMRC) in EDTA tubes from a human donor" - what are these blood samples and collected for what? Who are the patients? 4) State the purpose of each of the experiments you are doing.  

Results- Difficult to understand because there is no specific research question. 

Discussion- 1) You need to discuss your results in a meaningful way and connect the different experiments. 2) Discuss other studies in the literature. 3) What are the limitations of the study? 4) The limitations of the study should be at the end of the discussion and not in the conclusion. The abstract conclusions, as stated earlier, are not correct. 

References

Search the literature. These studies are related to your work

Sordo-Bahamonde C, Lorenzo-Herrero S, Granda-Díaz R, Martínez-Pérez A, Aguilar-García C, Rodrigo JP, García-Pedrero JM, Gonzalez S. Beyond the anti-PD-1/PD-L1 era: promising role of the BTLA/HVEM axis as a future target for cancer immunotherapy. Mol Cancer. 2023 Aug 30;22(1):142. doi: 10.1186/s12943-023-01845-4. PMID: 37649037; PMCID: PMC10466776.

Sordo-Bahamonde C, Lorenzo-Herrero S, Gonzalez-Rodriguez AP, R Payer Á, González-García E, López-Soto A, Gonzalez S. BTLA/HVEM Axis Induces NK Cell Immunosuppression and Poor Outcome in Chronic Lymphocytic Leukemia. Cancers (Basel). 2021 Apr 7;13(8):1766. doi: 10.3390/cancers13081766. PMID: 33917094; PMCID: PMC8067870.

Demerlé C, Gorvel L, Olive D. BTLA-HVEM Couple in Health and Diseases: Insights for Immunotherapy in Lung Cancer. Front Oncol. 2021 Aug 31;11:682007. doi: 10.3389/fonc.2021.682007. PMID: 34532285; PMCID: PMC8438526.

Šedý JR, Ramezani-Rad P. HVEM network signaling in cancer. Adv Cancer Res. 2019;142:145-186. doi: 10.1016/bs.acr.2019.01.004. Epub 2019 Feb 27. PMID: 30885361.

Comments on the Quality of English Language

Thank you for inviting me to review the above-titled manuscript. The topic is interesting. However, there are problems in the manuscript.

Abstract- 1) I'm not sure why these numbers? Is this the journal style? 2) What is the research question? 3) What are you trying to prove? 4) Methods- What are you testing? 5) what samples? 6) Are these in-vitro or in-vivo studies? What is the study design? How did you assess your data? 7) Results cannot prove a cause-and-effect. It can be an association. No strong evidence for your conclusions. The last line in the conclusion should be omitted.

Introduction- 1) The introduction should be focused and show related studies. 2) What is the research question? What are you trying to prove? 

Methods- 1) Needs citations and references in different sections. What are the references for the method? 2) What is the validity and reliability? 3) The "newly diagnosed patients"- Did you start them on treatment? or the research started before their treatment? 3) "Blood samples were collected by the Roaya Unit at King Fahd Medical Research Cen-102 ter (KFMRC) in EDTA tubes from a human donor" - what are these blood samples and collected for what? Who are the patients? 4) State the purpose of each of the experiments you are doing.  

Results- Difficult to understand because there is no specific research question. 

Discussion- 1) You need to discuss your results in a meaningful way and connect the different experiments. 2) Discuss other studies in the literature. 3) What are the limitations of the study? 4) The limitations of the study should be at the end of the discussion and not in the conclusion. The abstract conclusions, as stated earlier, are not correct. 

References

Search the literature. These studies are related to your work

Sordo-Bahamonde C, Lorenzo-Herrero S, Granda-Díaz R, Martínez-Pérez A, Aguilar-García C, Rodrigo JP, García-Pedrero JM, Gonzalez S. Beyond the anti-PD-1/PD-L1 era: promising role of the BTLA/HVEM axis as a future target for cancer immunotherapy. Mol Cancer. 2023 Aug 30;22(1):142. doi: 10.1186/s12943-023-01845-4. PMID: 37649037; PMCID: PMC10466776.

Sordo-Bahamonde C, Lorenzo-Herrero S, Gonzalez-Rodriguez AP, R Payer Á, González-García E, López-Soto A, Gonzalez S. BTLA/HVEM Axis Induces NK Cell Immunosuppression and Poor Outcome in Chronic Lymphocytic Leukemia. Cancers (Basel). 2021 Apr 7;13(8):1766. doi: 10.3390/cancers13081766. PMID: 33917094; PMCID: PMC8067870.

Demerlé C, Gorvel L, Olive D. BTLA-HVEM Couple in Health and Diseases: Insights for Immunotherapy in Lung Cancer. Front Oncol. 2021 Aug 31;11:682007. doi: 10.3389/fonc.2021.682007. PMID: 34532285; PMCID: PMC8438526.

Šedý JR, Ramezani-Rad P. HVEM network signaling in cancer. Adv Cancer Res. 2019;142:145-186. doi: 10.1016/bs.acr.2019.01.004. Epub 2019 Feb 27. PMID: 30885361.

Author Response

We would like to take this opportunity to thank the reviewers for their time and effort in reviewing this article. All their comments are valuable and have been taken into consideration for improving the manuscript. Regarding the English language, the manuscript has been sent and revised by a native English speaker. Below are our responses point by point.

Abstract-

1) I'm not sure why these numbers? Is this the journal style? 2) What is the research question? 3) What are you trying to prove? 4) Methods- What are you testing? 5) what samples? 6) Are these in-vitro or in-vivo studies? What is the study design? How did you assess your data? 7) Results cannot prove a cause-and-effect. It can be an association. No strong evidence for your conclusions. The last line in the conclusion should be omitted.

1) I'm not sure why these numbers? Is this the journal style?

Answer: Agreed, numbers have been removed from the Abstract.

2) What is the research question?

Answer: Agreed, the aim of the study is now clearly stated: “This study aims to evaluate the effects of blocking HVEM expressed by tumor cells on the proliferation of CD4+ T cells in vitro and measure the expression of HVEM gene expression in peripheral blood mononuclear cells (PBMCs) collected from Acute Lymphocytic Leukemia (ALL) patients and healthy control.” Page 1, lines 24-27

3) What are you trying to prove?

Answer: In the abstract, the aim of the study is now clearly stated: “This study aims to evaluate the effects of blocking HVEM expressed by tumor cells on the proliferation of CD4+ T cells in vitro and measure the expression of HVEM gene expression in peripheral blood mononuclear cells (PBMCs) collected from Acute Lymphocytic Leukemia (ALL) patients and healthy control.” Page 1, lines 24-27

4) Methods- What are you testing?

Answer: In the abstract, the method was rewritten very briefly as we have limited words according to the journal instruction “HVEM gene expression in tumor cell lines and PBMC from ALL and healthy controls was measured by RT-qPCR. Tumor cells were left untreated (control) or treated with an HVEM blocker before co-culturing with CD4+T cells in vitro in the CFSE-dependent proliferation assay.” Page 1, lines 27-30

5) what samples?

Answer: The confusion that comes from using the word “sample” is very true. Therefore, this word has been replaced with the type of sample used in the whole manuscript. For example, “HVEM gene expression in tumor cell lines and PBMC from ALL and healthy controls was measured by RT-qPCR. Tumor cells were left untreated (control) or treated with an HVEM blocker before co-culturing with CD4+T cells in vitro in the CFSE-dependent proliferation assay.” Page 1, lines 27-30

6) Are these in-vitro or in-vivo studies? What is the study design? How did you assess your data?

Answer: Now under the abstract we stated that “This study aims to evaluate the effects of blocking HVEM expressed by tumor cells on the proliferation of CD4+ T cells in vitro and measure the expression of HVEM gene expression in peripheral blood mononuclear cells (PBMCs) collected from Acute Lymphocytic Leukemia (ALL) patients and healthy control.”, However, due to the word number limit allowed in the abstract, we could not elaborate more in the study design nor the statistics used to assess the results. Significant results are now included in the abstract. Page 1, lines 24-37

7) Results cannot prove a cause-and-effect

Answer: Agreed; there has been a great reduction in the abstract, and the ROC analysis has been omitted to fit the number of words. The abstract now included ROC analysis results that may prove cause-and-effect. Page 1, lines 34-36

Introduction-

1) The introduction should be focused and show related studies.

Answer: We have rewritten the whole introduction using various HVEM articles and the suggested literature. Pages 1-2, lines 41-94

2) What is the research question? What are you trying to prove? 

Answer: At the end of the introduction, we mentioned the scientific gap with this study's aim and objectives: "The use of HVEM blockade in leukemic samples has yet to be investigated. The clinical significance of HVEM as an immune checkpoint target with a potential prognostic marker in malignant tumors has been hypothesized in various cancers but not in ALL. Therefore, this study aims to evaluate the effects of blocking HVEM expressed by tumor cells on the proliferation of CD4+ T cells in vitro after measuring the expression of HVEM on multiple adherent and non-adherent tumor cell lines and examine if HVEM could be used as a prognostic biomarker in ALL.” Page 2, lines 88-94

Methods-

1) Needs citations and references in different sections. What are the references for the method? 2) What is validity and reliability? 3) The "newly diagnosed patients"- Did you start them on treatment? or the research started before their treatment? 3) "Blood samples were collected by the Roaya Unit at King Fahd Medical Research Cen-102 ter (KFMRC) in EDTA tubes from a human donor" - what are these blood samples and collected for what? Who are the patients? 4) State the purpose of each of the experiments you are doing.  

  • Needs citations and references in different sections.

Answer: Agreed, now we have cited 8 References in the methodology (highlighted in yellow). Page 3-4, lines 102,104, 112,118, 125, 135, 146, 173

  • What is validity and reliability?

Answer: Each experiment has from 2 to 3 repeats which is indicated in each figure legend. In addition, statistical analysis used in written in the method section “IBM SPSS Statistics 29.0.1.0 software and GraphPad Prism 10.0.3 (GraphPad Software, Inc.) were used for statistical analyses of relative gene expression of HVEM, where P0.05 was considered statistically significant. Non-parametric tests were chosen based on whether the included variables are categorical or have a skewed distribution accordingly. Significant changes in gene expression between non-malignant controls and ALL samples were noted using an unpaired two-tailed t-test. Additionally, one-way ANOVA (two-tailed Mann-Whitney and Kruskal-Wallis tests) was applied to select parameters to compare the three groups. Results are presented as the mean ± standard error of the mean (SEM). To examine the sensitivity and specificity of HVEM as a possible biomarker, receiver operating characteristic (ROC) curves were plotted using their expression values (2-ΔΔCq) in malignant ALL and non-malignant controls. P≤0.05 was considered to indicate a statistically significant difference.” Page 4, lines 175-186

  • The "newly diagnosed patients"- Did you start them on treatment? or the research started before their treatment?

Answer: Agreed, there has been confusion; we have stated, “ALL blood samples were collected from 1. newly diagnosed patients before starting treatment, 2. patients in remission phase; and 3. those patients experiencing Relapse/Refractory phase.” Page 3, lines 141-143

  • "Blood samples were collected by the Roaya Unit at King Fahd Medical Research Cen-102 ter (KFMRC) in EDTA tubes from a human donor" - what are these blood samples and collected for what? Who are the patients?

Answer: Agreed, it was not clear. Now we have written the purpose of each experiment at the beginning of each method. Clarify this sentence specifically as stated “To examine the proliferation of CD4+ T cells in response to HVEM+ tumor cells, CD4+ T cells were isolated from the peripheral blood of healthy individuals by the Roaya Unit at King Fahd Medical Research Center (KFMRC).” Page 3, lines 109-111

  • State the purpose of each of the experiments you are doing.  

Answer: Agreed; at the beginning of each method, we have stated a sentence including the purpose of the experiment. For example, we stated, “To examine the effect of HVEM-expressing tumor cells on T cell proliferation in vitro, it is important to measure the expression of HVEM in different tumor cell lines.” Page 3, lines 97-98. The same was done with each experiment in the whole method highlighted in yellow. Pages 2-4

Results-

Difficult to understand because there is no specific research question. 

Answer: Agreed, as the aim is now stated in the abstract, the end of the discussion, and the purpose of each experiment is written, the whole result section was revised and rewritten. All yellow highlights indicate changes done for more clarification. Pages 4-11

Discussion-

  • You need to discuss your results in a meaningful way and connect the different experiments.

Answer: Agreed, we have discussed our results and connected different experiments. For example, as stated “Herein, we evaluated the therapeutic target of HVEM on cancer, reporting that HVEM expression is significantly upregulated in leukemic cell line in vitro at gene and protein levels. Higher HVEM mRNA expression was observed in non-adherent leukemic cells compared to adherent breast cancer and Hepatocellular carcinomas and healthy human embryonic kidney (Hek 293) fitting previous studies that showed HVEM is expressed by A20 leukemia cells (35).” The same was done when discussing all our results. Pages 11-13

  • Discuss other studies in the literature.

Answer: Agreed, we have used the suggested articles related to the same research scope and linked them in the discussion. The whole discussion has been rewritten. Pages 11-13

3) What are the limitations of the study?

Answer: The limitations were in the conclusion section but have been moved to the end of the discussion section. The limitations statement is “Consistent with published literature, the current study has some limitations; ALL samples were difficult to obtain with the three study criteria: newly diagnosed, remission, relapse, and refractory. The number of collected samples is 33, with 23 patient samples and 10 healthy controls. The mean age ± SEM of healthy control subjects was 35 ± 3 compared to 8 ± 1.3 of ALL patients. Collecting blood samples from healthy pediatric patients not admitted to hospitals is a constraint. In the case of collecting samples with hospital-admitted pediatric samples, other underlying health conditions could not be excluded. Future ongoing studies are underway in our laboratory to delineate the HVEM mechanism through molecular silencing of the HVEM gene and examine the effect on T-cell responses. In addition, performing a phenotypic analysis of PBMCs may provide insights into using HVEM as a biomarker in ALL. The serum levels of HVEM will be tested and compared to the HVEM gene levels.” Page 13, lines 516-527

4) The limitations of the study should be at the end of the discussion and not in the conclusion. The abstract conclusions, as stated earlier, are not correct. 

Answer: Agreed, the limitation has been moved to the end of the discussion and the abstract conclusion has been changed. “These results indicate that HVEM is an inhibitory molecule that may serve as a potential biomarker and target for immunotherapy.” At the end of the discussion, Page 13, lines 516-527, At the end of the Abstract, Page 1, lines 36-37

References

Search the literature. These studies are related to your work.

  • Sordo-Bahamonde C, Lorenzo-Herrero S, Granda-Díaz R, Martínez-Pérez A, Aguilar-García C, Rodrigo JP, García-Pedrero JM, Gonzalez S. Beyond the anti-PD-1/PD-L1 era: promising role of the BTLA/HVEM axis as a future target for cancer immunotherapy. Mol Cancer. 2023 Aug 30;22(1):142. doi: 10.1186/s12943-023-01845-4. PMID: 37649037; PMCID: PMC10466776. 
  • Sordo-Bahamonde C, Lorenzo-Herrero S, Gonzalez-Rodriguez AP, R Payer Á, González-García E, López-Soto A, Gonzalez S. BTLA/HVEM Axis Induces NK Cell Immunosuppression and Poor Outcome in Chronic Lymphocytic Leukemia. Cancers (Basel). 2021 Apr 7;13(8):1766. doi: 10.3390/cancers13081766. PMID: 33917094; PMCID: PMC8067870.
  • Demerlé C, Gorvel L, Olive D. BTLA-HVEM Couple in Health and Diseases: Insights for Immunotherapy in Lung Cancer. Front Oncol. 2021 Aug 31;11:682007. doi: 10.3389/fonc.2021.682007. PMID: 34532285; PMCID: PMC8438526.
  • Šedý JR, Ramezani-Rad P. HVEM network signaling in cancer. Adv Cancer Res. 2019;142:145-186. doi: 10.1016/bs.acr.2019.01.004. Epub 2019 Feb 27. PMID: 30885361.

Answer: Thanks for providing these references which helped a lot in rewriting the introduction and discussion now have included in the manuscript. Pages 13-14, Reference numbers 1; line 551, Reference 10; line 571, Reference numbers 15; line 584, and Reference numbers 19; line 594.  

Reviewer 2 Report

Comments and Suggestions for Authors

This work deals with the expression of HVEM in some cell lines (K562, MCF7, HepG2 and Hek293). The effect of this molecule on CD4+ T cell proliferation and the expression of HVEM on ALL.

I think that the authors do not focus their study on a specific point.

Indeed, they start looking at the expression of HVEM on cells of different histotypes without any clear reason. After, they look at the inhibitory effect of this molecule on CD4+ cells, and finally they analysed a cohort of patient with ALL to point out that HVEM is highly expressed on these cells and there is a correlation with the prognosis. 

For all these points, the data shown are limited, highly expected if not already well known and the performed experiments not enough to give a clear answer to a specific question. Indeed, that HVEM is widely expressed on different cell types is known, the ligands are known, and the effects of blocking it are known. On the other hand, the authors did not analyse the ligands of this molecule or a specific molecule-molecule interaction.

The images of figure 1 are not relevant for the message of the manuscript. Also, immunofluorescence and microscopy analysis could give some information on the expression of HVEM on target cells, but the authors performed PCR that give information on mRNA not on actual protein expression. In addition, figure one is characterised by a strong light/strong brown background. These images should be substituted with better ones.

The isolation of CD4+ T cells is not good. There are several kits described in the literature where one can get really purified T cell subsets. All the data shown in figure 2 could be shown in a supplementary figure instead. The FACS dot plot seems not well compensate also. Further, it would be better to use antibody labelled with PE and/or APC (alexafluor647) instead of double conjugates such as Pe-Cy7 and APC-Cy7. Also, the use of 7AAD together of the other two can be tricky, having spectra partly overlapping although distinguishable.  

The blocking experiments on proliferation of CD4+ T cells have been analysed improperly. There are some software that can make a good analysis of CFSE reduction as an indicator of proliferation, such as ModFit and also FlowJo. The authors should use one of these software packages. However, from the plots shown is not evident at all the presence of proliferating CD4+ T cells. No CD4+ T cells with a larger size (like proliferating cells) are present in the FACS plots.

The blocking experiments have been performed at a very high concentration of the peptide. Also, CD4+ T cells have been added to six well plates, and it is well known that T cells are not in good condition if plated in flat-bottomed plates and at a quite low cell density. Finally, the authors tested proliferation after 72h, referring to a paper that it is not easy to get. Also, it is well known that T cells proliferate strongly and in a short time period only if they are stimulated with anti-CD3 and anti-CD28 antibodies together (usually associated with beads or linked to the plastic). I would say that it is almost impossible that CD4+ T cells can proliferate in the experimental conditions applied.

The cohort of patients and healthy control have been analysed for the mRNA expression of HVEM, but not for the protein expression. Thus, these findings are limited and incomplete. The authors should show FACS histograms for the protein expression of HVEM.

Overall, the manuscript is dealing with an interesting topic, but it appears not well-organised, presented and experiments performed are not feasible/credible.

Comments on the Quality of English Language

English editing is needed.

Author Response

We would like to take this opportunity to thank the reviewers for their time and effort in reviewing this article. All their comments are valuable and have been taken into consideration for improving the manuscript. Regarding the English language, the manuscript has been sent and revised by a native English speaker. Below are our responses point by point.

I think that the authors do not focus their study on a specific point.

Answer:  Agreed, the aim of the study is now clearly stated: “This study aims to evaluate the effects of blocking HVEM expressed by tumor cells on the proliferation of CD4+ T cells in vitro and measure the expression of HVEM gene expression in peripheral blood mononuclear cells (PBMCs) collected from Acute Lymphocytic Leukemia (ALL) patients and healthy control.” Page 1, lines 24-27

Indeed, they start looking at the expression of HVEM on cells of different histotype without any clear reason. After, they looked at the inhibitory effect of this molecule on CD4+ cells, and finally they analysed a cohort of patients with ALL to point out that HVEM is highly expressed in these cells and there is a correlation with the prognosis. For all these points, the data shown are limited, highly expected if not already well known and the performed experiments not enough to give a clear answer to a specific question. Indeed, that HVEM is widely expressed in different cell types is known, the ligands are known, and the effects of blocking it are known. On the other hand, the authors did not analyse the ligands of this molecule or a specific molecule-molecule interaction.

Answer:  We agreed that there are many in vitro and in vivo studies on HVEM. However, there have also been controversies on HVEM in cancer, as stated in the introduction: “On the other hand, HVEM has also been shown to enhance tumor regression and anti-tumor responses. Investigations have demonstrated that HVEM overexpression increases the survival incidence of pancreatic and bladder tumors (19–21). Chemotherapy with adoptive cells using CD8+ BTLA+ TILs is linked to better clinical outcomes for managing melanoma through an increased response to IL-2. The ligation of BTLA with HVEM has activated the Akt/PKB pathway thus preventing apoptosis of CD8+BTLA+ TILs (22).” Page 2, lines 77-83

In addition, we have written the scientific gap at the end of the introduction as stated “The use of HVEM blockade in leukemic samples has yet to be investigated. The clinical significance of HVEM as an immune checkpoint target with a potential prognostic marker in malignant tumors has been hypothesized in various cancers but not in ALL. Therefore, this study aims to evaluate the effects of blocking HVEM expressed by tumor cells on the proliferation of CD4+ T cells in vitro after measuring the expression of HVEM on multiple adherent and non-adherent tumor cell lines and examine if HVEM could be used as a prognostic biomarker in ALL.” Page 2, lines 88-94

Alia Aldahlawi; Fatemah Basingab; Jehan Alrahimi; Kawther Zaher; Peter Natesan Pushparaj; Mohammed A Hassan; Kaltoom Al-Sakkaf Herpesvirus Entry Mediator as a Potential Biomarker in Breast Cancer Compared with Conventional Cytotoxic T‑lymphocyte-associated Antigen 4. Biomed Rep 2023, 19, 56.

The images in Figure 1 are not relevant to the message of the manuscript. Also, immunofluorescence and microscopy analysis could give some information on the expression of HVEM on target cells, but the authors performed PCR that gives information on mRNA not on actual protein expression. In addition, figure one is characterized by a strong light/strong brown background. These images should be substituted with better ones.

 Answer: Agreed; based on these comments, we have stained and analyzed all tumor cell lines with anti-HVEM-PE mAbs. The results of staining are included in the result section (Figure 1). In addition, all figures with a strong light/strong brown background have been modified. Pages 5-6, lines 206- 213, 260-289

The isolation of CD4+ T cells is not good. There are several kits described in the literature where one can get really purified T cell subsets.

Answer: For CD4+ T cell isolation, MACS separation beads were used according to the manufacturing instructions. We agree that we have not reached above 90% purity; however, based on a paper we used after getting almost 80% purity that the MACS residuals after MACS separation would be 18% CD8 T cells and less than 02% CD4- and CD8- (Figure 5 on the paper below). Therefore, we accepted this CD4+ purity yield and proceeded with the CFSE proliferation assay.

Article Source: An Innovative Cascade System for Simultaneous Separation of Multiple Cell Types
Pierzchalski A, Mittag A, Bocsi J, Tarnok A (2013) An Innovative Cascade System for Simultaneous Separation of Multiple Cell Types. PLOS ONE 8(9): e74745. https://doi.org/10.1371/journal.pone.0074745

All the data shown in Figure 2 could be shown in a supplementary figure instead. The FACS dot plot seems not well compensated also.

Answer: Agreed, Figure 2 has been removed from the main manuscript to supplementary as suggested.

Further, it would be better to use antibodies labelled with PE and/or APC (alexafluor647) instead of double conjugates such as Pe-Cy7 and APC-Cy7. Also, the use of 7AAD together with the other two can be tricky, having spectra partly overlapping although distinguishable.  

Answer: Yes, it is common to face some difficulty in separating signals when using PE-Cy7 and APC-Cy7 together because of the similar emission spectra, however, we have compensated all fluorochrome using a single stained fluorochrome and fluorescent minus one before running the samples. In addition, unstained cells were also used. Our reference is the paper below that uses similar fluorochrome and the same flow device. In this article, they stained in one tube with CD4-P-Cy7 and APC-H7 (allophycocyanin-Hilite®7-BD). APC-H7 (allophycocyanin-Hilite®7-BD) and its analog APC-Cy7 (allophycocyanin-cyanine 7) are APC-tandem dyes that exhibit similar spectral properties with maximum absorption at ∼650 nm.

Article Source: An Innovative Cascade System for Simultaneous Separation of Multiple Cell Types
Pierzchalski A, Mittag A, Bocsi J, Tarnok A (2013) An Innovative Cascade System for Simultaneous Separation of Multiple Cell Types. PLOS ONE 8(9): e74745. https://doi.org/10.1371/journal.pone.0074745

We have also used these fluorochrome combinations in our previously published paper but will consider this important comment in our future work.

Alia Aldahlawi; Afnan Alqadiri; Hadil Alahdal; Kalthoom Al-Sakkaf; Jehan Alrahimi; Fatemah Basingab Tumor Necrosis Factor Alpha and Lipopolysaccharides Synergistic Effects on T-Cell Immunoglobulin and Mucin Domain 3 Regulation in Dendritic Cells. J King Saud Univ Sci 2022, 34.

The blocking experiments on proliferation of CD4+ T cells have been analysed improperly. There are some software that can make a good analysis of CFSE reduction as an indicator of proliferation, such as ModFit and also FlowJo. The authors should use one of these software packages. However, from the plots shown is not evident at all the presence of proliferating CD4+ T cells. No CD4+ T cells with a larger size (like proliferating cells) are present in the FACS plots.

Answer: FlowJo software was used in the analysis of CFSE after compensation “Data were then analyzed using FlowJoTM version 10 software (Becton Dickinson, USA).” Page 3, line 136

The results of the blocking experiment have been rewritten; all changes made are in yellow highlights.  CD4+ T cells have divided 5 times evidenced by the peak of each division. We have used the percentage of the last division for comparison between different treatments. We have added in the text a comparison with non-proliferated CD4+ T cells for more clarity of the figure. Pages 5-6, lines 219- 258

The blocking experiments have been performed at a very high concentration of peptide.

Answer: Agreed, we have used 20ng/100ul for 100000 tumor cells, which has been mistakenly written as a microgram. In addition, blocking HVEM in tumor cells was performed for 2 hours in a falcon tube in the fridge before being washed and plated with T cells. All these have been clarified in the method section of the manuscript (Page 3, line 120).

Also, CD4+ T cells have been added to six-well plates, and it is well known that T cells are not in good condition if plated in flat-bottomed plates and at a quite low cell density. Finally, the authors tested proliferation after 72h, referring to a paper that it is not easy to get. Also, it is well known that T cells proliferate strongly and in a short time period only if they are stimulated with anti-CD3 and anti-CD28 antibodies together (usually associated with beads or linked to the plastic). I would say that it is almost impossible that CD4+ T cells can proliferate in the experimental conditions applied.

Answer: Agreed that T cells will be always better when close to each other, and a round bottom is better than a flat bottom. In our study, we used the paper below as a reference in which they reached 20-fold expansion after 4 days of incubation using 6 well plates. We have plated 1 million CD4+ T cells in one well with 100,000 tumor cells at a ratio of 10:1, so the number of T cells added 10 times more than 96 well plates to keep CD4+ T cells together. References below are included in the method-related

  • Lewis MD, de Leenheer E, Fishman S, Siew LK, Gross G, Wong FS. A reproducible method for the expansion of mouse CD8+ T lymphocytes. J Immunol Methods. 2015 Feb;417:134-138. doi: 10.1016/j.jim.2015.01.004. Epub 2015 Jan 17. PMID: 25602136; PMCID: PMC4352898.
  • Basingab FS, Ahmadi M, Morgan DJ. IFNγ-Dependent Interactions between ICAM-1 and LFA-1 Counteract Prostaglandin E2-Mediated Inhibition of Antitumor CTL Responses. Cancer Immunol Res. 2016 May;4(5):400-11. doi: 10.1158/2326-6066.CIR-15-0146. Epub 2016 Feb 29. PMID: 26928462.

The cohort of patients and healthy control have been analysed for the mRNA expression of HVEM, but not for the protein expression. Thus, these findings are limited and incomplete. The authors should show FACS histograms for the protein expression of HVEM.

Answer: As the main aim of this study is to examine HVEM gene expression as a potential biomarker in the PBMCs of ALL patients, we did not look at the surface protein. From our previous study in breast cancer, the serum levels of HVEM are high suggesting that HVEM might be shed from the cell surfaces. Therefore, we have suggested the measurement of serum HVEM in correlation to HVEM gene expression in our future directions. Page 13, line 527

Alia Aldahlawi; Fatemah Basingab; Jehan Alrahimi; Kawther Zaher; Peter Natesan Pushparaj; Mohammed A Hassan; Kaltoom Al-Sakkaf Herpesvirus Entry Mediator as a Potential Biomarker in Breast Cancer Compared with Conventional Cytotoxic T‑lymphocyte-associated Antigen 4. Biomed Rep 2023, 19, 56.

Overall, the manuscript is dealing with an interesting topic, but it appears not well-organized, presented and experiments performed are not feasible/credible.

Answer: We hope that by addressing all the above issues and making substantial changes, we presented the manuscript with a better and accepted version. Changes include rewriting the abstract with the aim clearly stated, a more focused introduction, references in addition to the methods used, re-analyzing some of the results, and writing the discussion in the context of the other literature.

Reviewer 3 Report

Comments and Suggestions for Authors

The paper is generally good, but the methodology is confusing, which needs to be corrected before it might be accepted for publication. 

1. Light microscopy images should be analyzed and presented is a way that shous the changes in cells after the co-cultures. 

2. Why the Authors performed the experiments in 2 replicates? Maybe it should be presented as prelinunary results and not the full article?

Otherwise the topic is very interesting thus I encourage them to revise the paper. 

Author Response

We would like to take this opportunity to thank the reviewers for their time and effort in reviewing this article. All their comments are valuable and have been taken into consideration for improving the manuscript. Regarding the English language, the manuscript has been sent and revised by a native English speaker. Below are our responses point by point.

The paper is generally good, but the methodology is confusing, which needs to be corrected before it might be accepted for publication. 

Answer: Agreed; at the beginning of each method, we have stated a sentence including the purpose of the experiment. For example, we stated, “To examine the effect of HVEM-expressing tumor cells on T cell proliferation in vitro, it is important to measure the expression of HVEM in different tumor cell lines.” Page 3, lines 97-98. The same was done with each experiment in the whole method highlighted in yellow. Pages 2-4

  1. Light microscopy images should be analyzed and presented is a way that shous the changes in cells after the co-cultures. 

Answer:  Agreed, in Figure 2 we have included the images of cells after the co-culture as stated “ HVEM+ K562 cells and HVEM- Hek239 were treated first with 20 ng/ 100 ul HVEM blocker using 1×105 cells for 2 hours before being washed and then co-cultured with CFSE-labelled CD4+ T cells at 1 tumor or healthy control cells to 10 CFSE-CD4+ T cells ratio. The results showed that HVEM+K562 tumor cells expanded normally when they were cultured alone or co-cultured with CFSE-CD4+ T cells without HVEM blocker treatment of tumor cells. In contrast, the number of tumor cells reduced significantly when HVEM+K562 was treated with the HVEM blocker before being co-cultured with CFSE-CD4+ T cells (Figure 2, left column). In Hek293 cultures, cells appeared similar when Hek293 cells were grown alone or with CFSE-CD4+ T cells post Hek293 cells treatment with HVEM blocker (Figure 2, right column). These results indicated that the HVEM blocker at 20 ng/ 100 ul concentration was not toxic and that the healthy control Hek293 cells grew normally after HVEM blocker treatment. Flow cytometric analysis has further investigated the reduction of HVEM+K562 cells.”. Page 5, lines 230- 242

  1. Why the Authors performed the experiments in 2 replicates? Maybe it should be presented as prelinunary results and not the full article?

Answer: Agreed. To clarify, Light microscopy images are taken routinely to check on the viability of cells before proceeding with any further experiments; however, the photos were taken from 2 wells each time, therefore corrected in the manuscript as stated, “Results are collected from 2 wells of 3 separate experiments.” Page 7, lines 289-290

Otherwise the topic is very interesting thus I encourage them to revise the paper.

Answer: The whole paper has been revised

Round 2

Reviewer 1 Report

Comments and Suggestions for Authors

Abstract- What is the research question? What are you trying to test or answer?

Introduction- last paragraph: State your research question and your hypotheses. 

Methods- Start with a section on "study design" that summarises your approach to answer your research questions and research approach or subsequent experiments.

Statistical analysis - add a reference

Discussion- Discuss the validity and reliability of experiments/studies. Discuss findings against other studies in the literature. 

Some statements need editing

Comments on the Quality of English Language

Abstract- What is the research question? What are you trying to test or answer?

Introduction- last paragraph: State your research question and your hypotheses. 

Methods- Start with a section on "study design" that summarises your approach to answer your research questions and research approach or subsequent experiments.

Statistical analysis - add a reference

Discussion- Discuss the validity and reliability of experiments/studies. Discuss findings against other studies in the literature. 

Some statements need editing

Author Response

Our sincere thanks to the reviewers for their valuable comments which we will do our best to address point by point. All changes are highlighted in yellow. Regarding the English language, the manuscript has been sent to the MDPI English language center, and a certificate of English accuracy was provided.

Comments and Suggestions for Authors

Abstract- What is the research question? What are you trying to test or answer?

Answer: Agreed, a research question is now included in the Abstract as stated: “This study aims to determine whether HVEM is an immune checkpoint target with inhibitory effects on anti-tumor CD4+ T cell responses in vitro and whether HVEM gene expression is dysregulated in patients with acute lymphocytic leukemia (ALL).” Page 1, lines 24-27

Introduction- last paragraph: State your research question and your hypotheses. 

Answer: Agreed; a research question is now stated at the end of the introduction with the hypothesis “Therefore, the aim of this study is to determine whether HVEM has an inhibitory effect on anti-tumor CD4+ T cell responses in vitro and whether HVEM gene expression is dysregulated in patients with ALL. We hypothesize that tumor-expressing HVEM can inhibit the proliferation of CD4+ T cells in vitro and that HVEM gene expression is a prognostic biomarker upregulated in ALL. Page 2, lines 100-104

Methods- Start with a section on "study design" that summarises your approach to answer your research questions and research approach or subsequent experiments.

Answer: Agreed, now we have added a section of study design at the beginning of the method used to answer the research question as stated

2.1 Study design

To determine whether HVEM has an inhibitory effect on anti-tumor CD4+ T cell responses in vitro, an experimental approach was employed in three stages. First, the expression of the HVEM gene and surface protein were measured in various tumor cell lines in vitro, resulting in the selection of tumor cell lines that expressed the highest HVEM. CD4+ T cells were then isolated from healthy donors. Last, the HVEM+ tumor cells were treated with HVEM blockade before being co-cultured with CD4+ T cells. The proliferation of CD4+ T cells in response to HVEM blockade-treated and untreated tumor cells was measured via CFSE-dependent assay.

To ascertain whether HVEM gene expression is dysregulated in patients with ALL, reverse transcription‑quantitative polymerase chain reaction (RT‑qPCR) was utilized on RNA isolated from malignant ALL patients and non-malignant healthy controls. Correlations between HVEM gene expression and clinicopathological data were conducted. Further, receiver operating characteristic (ROC) curve analysis was performed to evaluate the diagnostic ability of HVEM gene expression to discriminate between malignant ALL and non-malignant healthy controls.” Page 3, lines 106-121

Statistical analysis - add a reference (Add reference)

Answer: Agreed, now we have added a reference in the statistical analysis. Page 5, line 217

Discussion- Discuss the validity and reliability of experiments/studies. Discuss findings against other studies in the literature. 

Answer: Agreed; now we have added a paragraph in the discussion as stated “To ensure the validity of the conducted experiments, we employed rigorous experimental design, appropriate sampling and measurement tools, and statistical analysis. For the experimental design, as in previous studies, all experiments used a single variable and a control. Examples of the controls utilized in this study are Hek293 in vitro cells (4,41–43), HVEM untreated tumor cells (46), unstained CD4+ T cells (30), and non-malignant healthy controls (35). In addition, the replication of each experiment was conducted similarly to previous publications (30,35). Moreover, various programs were employed to analyze data, such as FlowJo for flow cytometric data and GraphPad Prism for statistical analysis (30,35). The results included in this study indicate whether the differences between groups are significant based on p-values (12). The consistency, stability, and repeatability of the experiment's results confirm the reliability of our study. ”  Page 14, lines 540-550

We have also mentioned that we used Hek293 as a control similar to a previous study that used the same control as part of validity (using the correct controls) as stated “Although Hek293 cells are immortalized human embryonic kidney cells, they were used as a healthy control in our study because they showed zero ∆∆Ct of HVEM gene expression. The low HVEM expression levels on Hek293 were noted previously since these cells were transfected with recombinant HVEM plasmid to induce the expression of HVEM. Hek293 cells were also used as healthy controls in previous studies (4,41–43).” Page 13, lines 466-470

We have also rearranged the discussion and linked each set of results with previous literature. Pages 12-15

Some statements need editing

Answer: Agreed; as English, is not our mother language, we have sent the paper to the MDPI English language center for further editing.

Reviewer 2 Report

Comments and Suggestions for Authors

Actually, the authors improved the manuscript, clarifying the aim and revising the manuscript markedly. 

Also, I remain on mind. Indeed, the rationale of studying adherent and non-adherent cells for the expression of HVEM is faint. Also, the specific analysis of CD4+ T cell proliferation is not justified in the context of ALL. 

Finally, the effect on proliferation of CD4+ T cells analysed with CFSE are not convincing. The gate on lymphocytes does not allow seeing activated T cells and at the same time some tiny cells/debris (with usually low CFSE content) have been considered for the anaslysis.

Comments on the Quality of English Language

English is good enough to understand the message.

Author Response

Our sincere thanks to the reviewers for their valuable comments which we will do our best to address point by point. All changes are highlighted in yellow. Regarding the English language, the manuscript has been sent to the MDPI English language center, and a certificate of English accuracy was provided.

Responses to Reviewer 2

Comments and Suggestions for Authors

Actually, the authors improved the manuscript, clarifying the aim and revising the manuscript markedly. Also, I remain on mind.

Indeed, the rationale of studying adherent and non-adherent cells for the expression of HVEM is faint.

Answer: Agreed. The reason for using adherent and non-adherent cells is now stated: “Variations in gene expression have been detected between an adherent and a non-adherent cell suspension of Hek293 after transcriptomic, genomic, and metabolic gene analysis, which are associated with cell membrane proteins (27). As HVEM is a surface protein, two adherent tumor cell lines, breast cancer (MCF-7) and hepatocellular carcinomas (HepG2), and non-adherent chronic myelogenous leukemia (CML) (K562) were also utilized. An embryonic immobilized kidney cell line (Hek293) served as a healthy control (28).” Page 3, lines 124-130.

In addition, we wanted to examine the expression of HVEM in vitro as a start that will help in choosing which type of cancer should we consider for patients’ samples. Moreover, we could not find a publication on HVEM expression in ALL patients compared to AML. For all these reasons the HVEM gene expression in various tumor cell lines in vitro was examined.

Also, the specific analysis of CD4+ T cell proliferation is not justified in the context of ALL. 

Answer: Based on some of the literature as stated in the manuscript “The transformation of chronic myeloid leukemia (CML) into acute lymphoid leukemia (ALL) has been reported in 20–30% of cancer patients during the terminal blast crisis stage, where more than 20% blasts are present in the patients’ peripheral blood and bone marrow (18).” Page 2, lines 70-73

The difficulty of growing ALL in vitro restricted the use of ALL cell lines in testing HVEM blockade in this current study. ALL blasts rely on their in vivo environment and undergo apoptosis ex vivo. Drug sensitivity testing in vitro with ALL is not commonly performed owing to the fast fall of ALL cells in these experiments, even when no anti-leukemic drugs are used (44). Mycoplasma contamination occurs in 5–10% of ALL cell lines. Moreover, 16–38% of ALL cell lines cross-contaminate with other ALL cell lines or lose their phenotypic features after several passages. ALL cell lines are not constantly in circulation, and they invade other tissues. As a result, supplying growth factors, adhesive substrates, and cytokines may result in treatment resistance (45) Based on the above reasons, CML (K562), which expresses the highest HVEM levels, was used in the HVEM blockade experiment.” Page 13, Lines 470-480

Finally, the effect on proliferation of CD4+ T cells analysed with CFSE are not convincing. The gate on lymphocytes does not allow seeing activated T cells and at the same time, some tiny cells/debris (with usually low CFSE content) have been considered for the anaslysis.

Answer: Agreed; we have re-analyzed the main data in FlowJo to exclude all the debris and re-gated in lymphocytes alone, as presented in Figure 3, page 9 in the manuscript. We got a better discrimination in which 20% of CFSE CD4+ T cells have reached the 5th division compared to HVEM blocker-untreated tumor cells, in which only 11% of CD4+ T cells reached the final division. Looking at the HVEM protein expression on K562 in Figure 1 page 7, 11 % of these tumor cells express HVEM protein, thus blocking them will get relatively more proliferation of CD4+ T cells. In addition, validation was conducted using the new after the re-gating mean of fluorescent intensity (MFI) and prism graph pad to indicate if the differences are significant which they are as we show in Figure 4 page 10.

Comments on the Quality of English Language

English is good enough to understand the message.

Answer: Agreed, as English is not our mother language, we have sent the paper to the MDPI English language center for further editing.

Reviewer 3 Report

Comments and Suggestions for Authors

The Authors have adressed all my issues and now I belive the paper is ready for publication.

Author Response

Our sincere thanks to the reviewers for their valuable comments which we will do our best to address point by point. All changes are highlighted in yellow. Regarding the English language, the manuscript has been sent to the MDPI English language center, and a certificate of English accuracy was provided.

Responses to Reviewer 3

Comments and Suggestions for Authors

The Authors have addressed all my issues and now I believe the paper is ready for publication.

Answer: Many thanks.